# Improved Bioavailability of Ebastine through Development of Transfersomal Oral Films

**DOI:** 10.3390/pharmaceutics13081315

**Published:** 2021-08-23

**Authors:** Nayyer Islam, Muhammad Irfan, Ameer Fawad Zahoor, Muhammad Shahid Iqbal, Haroon Khalid Syed, Ikram Ullah Khan, Akhtar Rasul, Salah-Ud-Din Khan, Alaa M. Alqahtani, Muzzamil Ikram, Muhammad Abdul Qayyum, Ahmed Khames, Sana Inam, Mohammed A. S. Abourehab

**Affiliations:** 1Department of Pharmaceutics, Faculty of Pharmaceutical Sciences, Government College University, Faisalabad 38000, Pakistan; nayyerislam1@gmail.com (N.I.); haroonkhalid80@gmail.com (H.K.S.); ikramglt@gmail.com (I.U.K.); akhtar.rasul@gcuf.edu.pk (A.R.); drsanainam@gmail.com (S.I.); 2Department of Chemistry, Government College University, Faisalabad 38000, Pakistan; fawad.zahoor@googlemail.com; 3Department of Clinical Pharmacy, College of Pharmacy, Prince Sattam bin Abdulaziz University, Alkharj 11942, Saudi Arabia; m.javed@psau.edu.sa; 4Department of Biochemistry, College of Medicine, Imam Mohammad Ibn Saud Islamic University (IM-SIU), Riyadh 11432, Saudi Arabia; sdikhan@imamu.edu.sa; 5Pharmaceutical Chemistry Department, College of Pharmacy, Umm Al-Qura University, Makkah 21955, Saudi Arabia; amqahtani@uqu.edu.sa; 6Department of Radiology, Madinah Teaching Hospital, The University of Faisalabad, Faisalabad 38000, Pakistan; drmuzzamilirfan89@gmail.com; 7Department of Chemistry, Division of Science & Technology, University of Education, Lahore 5600, Pakistan; hmaqayyum@ue.edu.pk; 8Department of Pharmaceutics and Industrial Pharmacy, Beni-Suef University, Beni-Suef 62514, Egypt; Ahmed.khamies@pharm.bsu.edu.eg; 9Department of Pharmaceutics and Industrial Pharmacy, Faculty of Pharmacy, Minia University, Minia 61519, Egypt; maabourehab@uqu.edu.sa; 10Department of Pharmaceutics, Faculty of Pharmacy, Umm Al-Qura University, Makkah 21955, Saudi Arabia

**Keywords:** bioavailability, ebastine, edge activator, in vivo, phospholipids, transfersomes

## Abstract

The main objective of this research work was the development and evaluation of transfersomes integrated oral films for the bioavailability enhancement of Ebastine (EBT) to treat allergic rhinitis. The flexible transfersomes, consisting of drug (EBT), lipid (Phosphatidylcholine) and edge activator (EA) Polyoxyethylene sorbitan monooleate or Sorbitan monolaurate, were prepared with the conventional thin film hydration method. The developed transfersomes were further integrated into oral films using the solvent casting method. Transfersomes were evaluated for their size distribution, surface charge, entrapment efficiency (EE%) and relative deformability, whereas the formulated oral films were characterized for weight, thickness, pH, folding endurance, tensile strength, % of elongation, degree of crystallinity, water content, content uniformity, in vitro drug release and ex vivo permeation, as well as in vivo pharmacokinetic and pharmacodynamics profile. The mean hydrodynamic diameter of transfersomes was detected to be 75.87 ± 0.55 nm with an average PDI and zeta potential of 0.089 ± 0.01 and 33.5 ± 0.39 mV, respectively. The highest deformability of transfersomes of 18.52 mg/s was observed in the VS-3 formulation. The average entrapment efficiency of the transfersomes was about 95.15 ± 1.4%. Transfersomal oral films were found smooth with an average weight, thickness and tensile strength of 174.72 ± 2.3 mg, 0.313 ± 0.03 mm and 36.4 ± 1.1 MPa, respectively. The folding endurance, pH and elongation were found 132 ± 1, 6.8 ± 0.2 and 10.03 ± 0.4%, respectively. The ex vivo permeability of EBT from formulation ETF-5 was found to be approximately 2.86 folds higher than the pure drug and 1.81 folds higher than plain film (i.e., without loaded transfersomes). The relative oral bioavailability of ETF-5 was 2.95- and 1.7-fold higher than that of EBT-suspension and plain film, respectively. In addition, ETF-5 suppressed the wheal and flare completely within 24 h. Based on the physicochemical considerations, as well as in vitro and in vivo characterizations, it is concluded that the highly flexible transfersomal oral films (TOFs) effectively improved the bioavailability and antihistamine activity of EBT.

## 1. Introduction

Ebastine (EBT), is a histamine H_1_ receptor blocker which is used to treat different allergic diseases [1]. The prevalence of allergic diseases is more than 20% of the global population [2]. It is estimated that 28.5% of European population is affected by allergic rhinitis every year [3]. The effectiveness of antihistaminic products varies considerably from one another [4]. To cope with allergic diseases, it is necessary to improve the efficiency of available treatment options. As far as EBT (BCS-II drug) is concerned, its oral bioavailability is about 40% [5,6]. To obtain maximum therapeutic benefits from EBT, its oral bioavailability should be enhanced [7]. For that purpose, several approaches have been employed by scientists to increase the bioavailability of EBT, such as complexation, nanoparticle formulation, fast disintegrating tablets and microemulsion, as well as spray drying technique [8,9,10,11]. Importantly, researchers are also exploring different ways to enhance absorptions of drug molecules to increase oral bioavailability. The low permeation of drugs is generally caused by mucosal barriers. The rationale of choosing transfersomes-based films was to cross these mucosal barriers. Thus, the enhanced and extended absorption of EBT from the oral films might increase the bioavailability. Therefore, transmucosal delivery of EBT through GIT could possibly be enhanced by encapsulating it into transfersomes [12].

Transfersomes are ultra-deformable carriers composed mainly of phospholipids and edge activators (EAs). The phospholipids amphiphilic molecules comprise a hydrophilic head and hydrophobic tail groups in their structure [13]. Besides, the EA preserves the integrity of transfersomes when going across small channels, as a result, maximizing stability thus deformity is procured [14]. It has been reported that nonionic surfactants, such as polyoxyethylene sorbitan monooleate (Tween 80^®^) and sorbitan monolaurate (Span 20^®^), are efficient edge activators (EAs) for phosphatidylcholine (PC) [15]. Transfersomes have been found suitable for the delivery of hydrophilic as well as hydrophobic therapeutic candidates through the intact membranes, thus increasing therapeutic delivery, owing to their flexible structure [16]. The highly deformable property of transfersomes allows them to penetrate through tight junctions and small pores present in membranes [17]. Similarly, phospholipids and EAs collectively confer the squeezing properties to the transfersomes [18].

Nevertheless, transfersomes in aqueous dispersion may create problems in the handling and storage of incorporated drugs. Fusion, leakage, hydrolysis and aggregation may limit the benefits of transfersomes as carriers [19]. However, stability can be provided to transfersomes by converting them into solid state. The characteristics of transfersomes could be possibly improved by integrating them into oral films. In addition, the oral absorption of low bioavailable drugs is reported to be increased by the administration of oral films [20,21,22]. Film-forming polymers, in the presence of other additives (e.g., plasticizers, penetration enhancer, flavors and sweeteners), play a critical role in the development of effective oral films. Previously, hydroxypropyl methylcellulose (HPMC-K15M) was investigated for the development of oral films [23,24]. The main reason for selecting HPMC is that it provides appropriate mechanical strength to the films [25,26,27]. The significance of current research work is that transfersomes-loaded oral films offer more absorptive properties than existing EBT formulations. Besides, transfersomes have the ability to diffuse through lipid membranes by modulating the membrane bilayer structure, acting as a permeation enhancer and making pathway to improve the absorption of drug molecules [28].

The objective of this work is to develop TOFs for the delivery of EBT through oral route to achieve improved therapeutic plasma levels. Moreover, the developed transfersomes and TOFs were evaluated for vesicle size distribution, polydispersity index, zeta potential, Fourier-transform infrared spectroscopy (FTIR), X-ray diffraction analysis (XRD), differential scanning calorimetry (DSC), scanning electron microscopy (SEM), in vitro drug release and ex vivo permeation, as well as the in vivo pharmacokinetic and pharmacodynamics parameters.

## 2. Materials and Methods

### 2.1. Materials

Phospholipon^®^ 90H (phosphatidylcholine; PC) was gifted by Lipoid GmbH (Ludwigshafen, Germany). Ebastine (EBT) was provided by SIMZ Pharmaceuticals (Lahore, Pakistan). Tween-80^®^ (Polyoxyethylene sorbitan monooleate), Span^-^20^®^ (Sorbitan monolaurate) and HPLC grade organic solvents, including chloroform (CHCl_3_) and methanol (MeOH), were generously donated by Wimits Pharmaceuticals (Lahore, Pakistan). Cellophane membrane with a molecular weight cut-off value of 12,000–14,000 KDa was purchased from Sigma-Aldrich Chemie GmbH (Taufkirchen, Germany). Hydroxypropyl methylcellulose (HPMC-K15M) having methoxy and hydroxypropyl contents of 19–24% and 7–12%, respectively, as well as glycerol and poly-sucralose, were purchased from Arsons Pharmaceuticals (Lahore, Pakistan). Phosphate buffered saline (PBS) with a pH of 7.4 was prepared freshly, sterile filtered and stored in the fridge for further use. Sterile filtered ultra-pure water (ELGA Lab water, High Wycombe, UK) was used for all the experiments.

### 2.2. Preparation of Transfersomes

Transfersomes (composition given in Table 1) were prepared with the conventional thin film hydration method, as described by Elkomy et al., with slight modifications [29]. Briefly, drug, PC and EAs were dissolved in chloroform/ methanol (2:1) mixture using a round bottle flask (Figure 1). The EBT (10%) was incorporated into the mass concentration of transfersomal formulations (Table 1). The mass concentration ratio of EBT to the total PC/EA concentration was 1:10. The organic solvents were then removed using a rotary evaporator (Rotavapor R-300, Buchi Labortechnik AG, Flawil, Switzerland) equipped with a vacuum pump at 50 °C. The developed film was hydrated with PBS (pH, 7.4) by thoroughly agitating and mixing through magnetic stirrer at room temperature for 1 h. Afterwards, the developed vesicles were sonicated for 30 min in a bath type sonicator (Elmasonic P, Elma Schmidbauer, Singen, Germany). Finally, the obtained transfersomes were further used in the development of TOFs after analysis.

### 2.3. Preparation of TOFs

The solvent casting method was used for the preparation of oral films with slight modifications [30]. The detailed formulations of TOFs are presented in Table 2. Briefly, the film solutions were prepared by dispersing the HPMC-K15M (2–3%) and glycerol (4–8%) in distilled water; subsequently, polysucralose (1%) was added. The mixture was stirred using a magnetic stirrer at 250 rpm and 25 °C for 12 h. The transfersomal dispersion containing 10 mg/mL of EBT was added to the film solution and subjected to homogenization using a Heidolph-4000 (Heidolph Instruments, Schwabach, Germany) for 5 min [31]. The solution was left over night to homogenize and degassed completely. The solution having a viscosity of 7250 mPa·s was casted with the help of a film applicator (BGD-219, Solvica Advanced Testing equipment, Düsseldorf, Germany) on a polyethylene terephthalate (PET) release liner Hostaphan-RN^®^, (Mitsubishi Polyester Film Incorporation, Greer, SC, USA) using a speed of 6 mm/s. Afterwards, the solvent was allowed to evaporate for 24 h at room temperature. Polyethylene monolayer CoTran™ (3M™ GmbH, Nordrhein-Westfalen, Germany) was used as backing layer. The films were peeled off and cut into desired sizes and were stored in a desiccator after wrapping in aluminum foil for further analysis [32].

### 2.4. Characterization of Transfersomes

Particle size distribution and ζ potential of the developed transfersomes were determined using dynamic light scattering Zetasizer Nano-ZS (Malvern Panalytica GmbH, Kassel, Germany) and laser doppler velocimetry (LDV), respectively. The instrument was equipped with 10 mW HeNe laser at a wavelength of 633 nm at 25 °C. The scattered light was detected at an angle of 173°. Measurement position and laser attenuation were automatically adjusted with each measurement. The disposable capillary cell (DTS1060, Malvern Instruments) was used for the measurements. The sample was always diluted with purified water (1:100) before each measurement. The refractive index of water (1.33) and viscosity (0.88 mPa·s) were taken into account for data analysis. The size distribution of vesicles for homogeneity or heterogeneity was determined from the polydispersity index (PDI). All samples were observed in triplicate with each measurement comprising of 15–100 runs, depending on the sample. Data are expressed as mean ± s.d.

### 2.5. Deformability Studies

In order to conduct deformability studies, the vesicle dispersion was passed through a polycarbonate membrane filter (Isopore™, Merck, MA, USA) with a pore size of 30 nm under constant pressure of 1.5 MPa of nitrogen stream for 15 min [33]. The transfersomal suspension was then collected in a container placed on a balance to note the weight change as a function of time.

### 2.6. Entrapment Efficiency (EE%)

The EE% of all the formulations was evaluated to analyze the impact of edge activators and lipids (PC) in drug entrapment [34]. The transfersomes were separated from the free drug by centrifugation at 10,000 rpm for 30 min and 4 °C using an Eppendorf centrifuge 4518 (Eppendorf, Hamburg, Germany). To remove the excess drug from the outer surface, the separated transfersomes were washed with PBS (pH, 7.4) three times. The free drug was measured using a Shimadzu-HPLC-LC-10AD (Shimadzu Corporation, Kyoto, Japan) at a wavelength of 257 nm. The EE % was calculated using the following equation [35]:(1)$$\mathrm{EE}\;(\%)=\frac{Q_{t}{-\;Q}_{u}}{Q_{t}\;}\times 100$$
where Q_t_ is the total quantity of drug added and Q_u_ is the quantity of free drug in the transfersomes.

### 2.7. Characterization of TOFs

#### 2.7.1. Viscosity Measurement of Film Solutions

A digital Brookfield^®^ Viscometer-DV2T (Brookfield Ametek, Middleborough, MA, USA) was coupled with a S-18 spindle to determine the viscosity of the TOF solution at a speed of 10 rpm. Triplicate readings of all film solutions were taken at 25 ± 2 °C [36].

#### 2.7.2. Film Thickness and Weight

The mean thickness of the film was measured at three different points using a standard size piece (2 × 2 cm^2^) for each film with the help of a digital micrometer Mitutoyo 500-196-30 (Mitutoyo Corporation, Kawasaki, Kanagawa, Japan). Besides, the average weight of the films was recorded using an analytical balance (Cubis-MSA-225P, Sartorius, Aubagne, France) [37].

#### 2.7.3. Surface pH

The surface of each film was moistened with distilled water in a petri dish for 1 min and pH was measured by placing an electrode of a Benchtop pH meter-S210 (Mettler-Toledo International Incorporation, Columbus, OH, USA) on the surface of the film. The pH for three independent films was measured and the average was determined. Data are expressed as mean ± s.d. [38].

#### 2.7.4. Folding Endurance

The endurance of films was considered as a measure of fold numbers. Films with size of 4 × 4 cm^2^ were repeatedly folded at the same point until broken. The number of folds of each film was recorded as endurance value [39].

#### 2.7.5. Tensile Strength and % of Elongation of TOFs

The tensile strength of films was measured by attaching films of 2 × 2 cm^2^ size with the grips of a tensile strength analyzer Shimadzu-AGS-100kNX (Shimadzu Corporation, Kyoto, Japan). The force required to move the grip away till the film ruptured was noted as tensile strength. Further, the % of elongation was also calculated [40].

#### 2.7.6. Moisture Content of TOFs

The weight of films before and after drying were recorded to calculate the water loss from oral films [41]. The film of approximately 2 × 2 cm^2^ in size was cut and weighed. This was nominated as initial weight (W_1_). Then, the same film was placed in a hot dry oven with temperature maintained at 50 ± 1 °C for 1 h. After taking it out of the oven, the dried film was weighed again and that was considered its final weight (W_2_). The water loss was calculated using following equation:(2)$$\mathrm{Water}\;\mathrm{loss}\;\left(\%\right)=\frac{W_{1}{-W}_{2}\;}{W_{1}}\times 100$$
where W_1_ is the initial weight of the film and W_2_ is the final weight of the film.

#### 2.7.7. Content Uniformity

The content uniformity of the oral films was determined using solvent extract techniques. Briefly, a film of 2 × 2 cm^2^ in size was dissolved in 20 mL of methanol. The solution was filtered through an MF-Millipore™ membrane filter having a pore size of 0.45 µm (Merck, Boston, MA, USA) to remove undissolved matter. Subsequently, different dilutions were prepared to be analyzed with a UV/VIS spectrophotometer (UV mini 1240, Shimadzu, Japan) at 254 nm wavelength. The content uniformity was then determined using calibration curve already constructed in the same solvent system with known EBT concentrations. The films without drug were taken as blank control [42].

#### 2.7.8. Reconstitution of TOFs

The double distilled water was added to reconstitute the films to confirm the colloidal solution. The oral film was re-dispersed in 1 mL of distilled water/buffer solution, pH of 6.8, following a previously published method [43]. The solution was stirred for 1 h. After reconstitution, the solution was analyzed for entrapment efficiency, particle size and deformability of transfersomes [44].

#### 2.7.9. Fourier-Transform Infrared Spectroscopy (FTIR)

The FTIR spectra of EBT, PC, Tween 80^®^, Span 20^®^ and the optimized formulation (i.e., ETF-5) were performed between 500 and 4000 cm^−1^ wavenumbers using an Agilent FTIR-Cary-360 (Agilent Scientific Instruments, Santa Clara, CA, USA) [45].

#### 2.7.10. Powder X-ray Diffraction (PXRD)

PXRD studies of pure drug and TOFs were conducted using a diffractometer, D8-advance (Bruker Corporation, Billerica, MA, USA). The samples were measured between the angles of 5° and 50° at 2θ [46].

#### 2.7.11. Differential Scanning Calorimetry (DSC)

The DSC analysis of pure drug, lipids and edge activators were performed using a simultaneous Thermal Analyzer SDT-Q600 (TA Instruments, New Castle, DE, USA), at a temperature scanning rate of 10 °C/min and nitrogen flow rate of 20 mL/min. The samples were run between 25 and 200 °C to perform DSC tests [47].

#### 2.7.12. Scanning Electron Microscopy (SEM)

In order to study the surface morphology of the films, the developed formulations were subjected to scanning electron microscopy using a Nova-Nano-SEM-450, (FEI, Hillsboro, OR, USA). The samples of pure drug and the prepared films were positioned on a copper grid and the surface texture was examined [46].

#### 2.7.13. Atomic Force Microscopy (AFM)

The surface texture of oral film was analyzed by the atomic force microscopy Dimension-XR SPM (Bruker Corporation, Billerica, MA, USA), using an etched silicon probe at tapping mode (i.e., intermittent contact mode). The dry film was placed on a glass stub and subjected to analysis. The oral film of 5 × 5 μm^2^ in size was examined at a 0.9 Hz scanning rate to measure its height from different points [48].

#### 2.7.14. In Vitro Drug Release

The USP paddles apparatus (Biobase Biodustry, Jinan, Shandong China) was used to study in vitro drug release from the developed TOFs. Briefly, an in vitro release study was performed in 0.1 N HCl (pH, 1.2) for 2 h and in phosphate buffer (pH, 6.8) for 24 h. Approximately 500 mL of 0.1 N HCl (pH, 1.2)/PBS (pH, 6.8) was poured in dissolution vessels. The film size was cut per unit dose and placed in a steel wire mesh. Then mesh was tied with paddles. The temperature was kept at 37 ± 0.5 °C and stirring speed was adjusted at 100 rpm. The samples (5 mL) were drawn at predetermined time intervals (0, 1, 2, 3, 4, 5, 6, 8 and 24 h) and fresh dissolution medium was added every time a sample was taken to maintain the sink conditions. The drug release was analyzed using Shimadzu-HPLC-LC-10AD (Shimadzu Corporation, Kyoto, Japan) at 257 nm [29]. The mobile phase was composed of an acetonitrile:methanol:ammonium acetate buffer (20:30:50) and was equipped with a C-18 column (5 mm, 25,034.0 mm; Agilent). The flow rate of 1 mL/min and column temperature of 40 °C were maintained throughout analysis.

#### 2.7.15. Ex Vivo Permeation

Ex vivo permeation of EBT from the optimized formulation ETF-5 was determined by vertical Franz diffusion cells. The Wistar rats were euthanized, their small intestines were separated and washed several times with normal saline. Subsequently, a small intestine was placed in PBS (pH, 7.4) at a temperature of 37 ± 1 °C. The ileum membrane of the small intestine was used as a permeation barrier with a permeation area of approximately 1.76 cm^2^. The receptor chamber was filled with 7 mL of PBS (pH, 7.4). The temperature of PBS was maintained at 37 ± 1 °C, whereas the stirring speed was adjusted to 100 rpm. At scheduled time points, a 0.2 mL sample was withdrawn from the receptor compartment using a micropipette. The amount of drug permeated through part of the small intestine was determined using Shimadzu-HPLC-LC-10AD (Shimadzu Corporation, Kyoto, Japan) equipped with a C-18 column (5 mm, 25,034.0 mm; Agilent) at 257 nm using an acetonitrile:methanol:ammonium acetate buffer (20:30:50) as mobile phase. The rate of permeation (i.e., flux) was calculated by plotting percentage of drug permeated per unit surface area against time [49]:(3)$${\mathrm{Steady}\;\mathrm{state}\;\mathrm{flux}\;(J}_{\mathrm{ss}}{)\;(µg/\mathrm{cm}}^{2}/h)=\frac{\mathrm{Permeation}\;\mathrm{rate}}{\mathrm{Per}\;\mathrm{unit}\;\mathrm{surface}\;\mathrm{area}}$$

#### 2.7.16. Compliance with Ethical Standards

All animal studies were approved by the Institutional Review Board and bioethical committee of the Government College University Faisalabad (Ref No. GCUF/ERC/2068, Study No. 19668, IRB No. 668, dated 5 September 2019). Wistar rats weighing 150–210 g aged 3–4 months were placed separately in steel cages filled with wood dust (replaced every 24 h). The rats were housed in a controlled environment (temperature, 22 ± 2 °C; humidity, 60 ± 10%, with an alternate 12 h light/dark cycle) for a week before and throughout all experiments. All the animals were provided free access to tap water and commercial laboratory feed ad libitum (Hi-Tech Feeds Pvt. Ltd., Lahore, Pakistan). Appropriate care was exercised to minimize the number of animal used as defined by the 3R principle, i.e., replacement, reduction and refinement. The study was carried out in strict compliance with the institutional guidelines and the recommendations of the “Guide for the care and use of laboratory animals, institute of laboratory animal resources”.

#### 2.7.17. In Vivo Pharmacokinetic Study

TOFs, plain film (loaded with crystalline EBT) and pure EBT suspension were evaluated for pharmacokinetic parameters in male Wistar rats. Randomly, the rats were divided in three groups (E, P and T), each having 6 rats (n = 6). Samples were administered orally to all the animals of the three groups. The EBT dose in each treatment was equivalent to 10 mg/kg body weight. The ether anesthesia was given to all rats for the placement of TOFs in the oral cavity. The oral films were cut into 4 × 4 mm^2^ pieces (each piece contained almost 2 mg of EBT) and placed in the oral cavity with the help of a Teflon spatula and forceps and subsequently swallowed with water. The films were intact when they were administered to the rats. Blood samples were withdrawn from the tail vein at predetermined time intervals (0.5, 1, 2, 3, 4, 5, 6, 8, 10, 12, 24, 48 and 72 h). Blood samples collected in heparinized plastic tubes were subjected to centrifugation (Centurion Scientific, Chichester, UK) at 1500 rpm for 5 min. The plasma was carefully pipetted out and stored in Eppendorf plastic tubes at −20 °C for further analysis.

#### 2.7.18. Extraction of EBT and Carebestine from Plasma

Equal quantities of plasma and absolute methanol were centrifuged at 10,000 rpm for 5 min. Nitrogen stream was applied (−40 °C) to dry the supernatant. The obtained residue was reconstituted with methanol and centrifuged at 10,000 rpm to collect the supernatant for further analysis using reverse phase isocratic HPLC Shimadzu, LC-10AD, (Shimadzu Corporation, Kyoto, Japan). 

#### 2.7.19. HPLC Analysis and Pharmacokinetic Parameters

The mobile phase was composed of acetonitrile:methanol:ammonium acetate buffer (20:30:50). To analyze EBT and carebastine in samples, the isocratic reverse phase Shimadzu-HPLC-LC-10AD (Shimadzu Corporation, Kyoto, Japan) equipped with a C-18 column (4.6 × 250 mm; Agilent) was run at 257 nm wavelength using flow rate of 1 mL/min. The temperature of the column was fixed at 40°C. The final concentrations of standard and sample (40 µg/mL) were prepared. After setting baseline, the samples were injected for elution. The Microsoft Excel PK Solver™ (version 2016) software was used to calculate maximum plasma concentration (C_max_), time to reach maximum plasma concentration (T_max_) and elimination half-life (K_e_), as well as the area under the plasma concentration–time curve (AUC_0–t_).

#### 2.7.20. In Vivo Pharmacodynamics (Histamine Induced Wheal and Flare Challenge)

Rats were divided into four groups each comprising of 6 rats (n = 6). The site for histamine-induced skin reaction was prepared. Histamine epi-cutaneous injection (0.05 mL of 100 µg/mL) was used to induce wheal and flare in all groups and the gap between the testing points was kept at 3 cm. The time points for wheal measurement were designated as 0, 4, 8 and 24 h. Single dose was administered to the rats in all groups except placebo. The area of wheal and flare was noted by tracing on a paper after 15 min of histamine injection. Diameter of wheal and flares was noted, based on an angle of 90°. The wheal and flare were measured at predetermined time intervals to evaluate the treatment efficiency of developed TOFs with reference to pure drug and the plain films. In addition, the animals were observed for any adverse effects during the study.
(4)$$\mathrm{Area}=\frac{{\pi \times D}_{L}{\times D}_{s}}{4}$$
(5)$${A=\pi \;(R)}^{2}$$
(6)$$\mathrm{Wheal}\;\mathrm{area}\;\mathrm{reduction}\;(\%)=100-\left(\;\frac{A_{t}}{A_{b}}\;\right)\times 100$$
where, D_L_ is the maximum wheal diameter and D_s_ is the minimum wheal diameter; A is the area of wheal and R is the radius of wheal; A_b_ is the area before treatment and (A_t_) is the area after treatment. 

### 2.8. Statistical Analysis

The SPSS^®^ software (version 21) was used to apply one-way ANOVA to in vitro drug release, ex vivo permeation and in vivo pharmacokinetic data, followed by a post-hoc Tukey’s test. A *p*-value < 0.05 was considered statistically significant for population difference among groups.

## 3. Results

### 3.1. Physicochemical Characterizations

The EBT loaded transfersomes were characterized to evaluate their physiochemical properties. The size of all formulations was found in the range from 66.3 ± 4.3 to 88.7 ± 2.8 nm with the polydispersity index ranging from 0.014 to 0.21. The surface charge of developed transfersomes was detected between −27.4 ± 2.14 and −37.7 ± 3.1 mV (Figure 2A). The EE (%) of all formulations was more than 90% (Figure 2B). The transfersomal formulation VS-3 showed the highest deformability value (18.52 mg/s); hence, it was selected, based on its high deformability value, to be integrated into oral films.

### 3.2. Characterization of TOFs

Table 3 shows characterization results of TOFs. The high rheology of film solution (from 6900 to 7400 mPa.s) was obtained from HPMC in the presence of the permeation enhancer and plasticizer for the development of oral films. TOFs appeared to be smooth, flexible and uniform. Films weight ranged from 157.4 ± 2.64 to 192.1 ± 1.15 mg and the average surface pH was approximately 6.8. The thickness of films was between 0.241 ± 0.03 and 0.387 ± 0.02 mm. ETF-5 formulation provided the best folding durability, with a tensile strength above 36.4 MPa. The elongation percentage of the films ranged between 4.1 ± 0.16 and 17.9 ± 0.31%. The content uniformity was found within specification (99–102%). Out of all developed film formulations, the ETF-5 film was chosen for further studies on the criterion of efficacious physiochemical presentation with minimal amounts of film-former and plasticizer.

### 3.3. Reconstitution of TOFs Containing Transfersomes

The EE % was found to be 94% after reconstitution of the optimized formulation ETF-5. The particle size and deformability were 68.8 ± 0.13 nm and 18.4 ± 0.11 mg/s, respectively. In addition, EE%, particle size and deformability were comparable with the original results. It supported that the properties of transfersomes remained intact upon incorporation into oral films.

### 3.4. Fourier-Transform Infrared Spectroscopy (FTIR)

The FTIR spectrum of pure EBT showed major peaks at 3050 cm^−1^, 2945 cm^−1^, 1452 cm^−1^, 1270 cm^−1^ and 1674 cm^−1^ wavenumbers, as shown in Figure 3. The phosphatidylcholine C=O stretching appeared at 1704 cm^−1^, due to the ester group. The phosphate stretch appeared at 1061 cm^−1^ and P=O vibration at 1251 cm^−1^. In addition, the Tween 80^®^ C=O stretch was seen at 1731 cm^−1^, C–H stretching at 2920 cm^−1^ and 2812 cm^−1^. In the Span 20^®^ IR spectra, the C=O stretch appeared at 1735 cm^−1^ and C–H stretch at 2842 cm^−1^ and 2924 cm^−1^. Overall, the FTIR spectra of TOF clearly matched with the main peaks of the pure drug. In ETF-5, the C–H stretching of ring, C–H stretching of (CH_3_), C=C stretching aromatic ring, C–N stretching and C=O stretching were found at same wavenumbers, parallel to pure EBT.

### 3.5. Morphological Evaluations Using SEM

The SEM results provided an irregular, rod-shaped and rough crystalline structure of the pure drug (Figure 4A). EBT crystals displayed length and width of varying sizes. The crystals had infinite shape with coarse surface. In comparison, the surface of TOFs was overall uniform, with minute uneven texture (Figure 4B). Further, small size transfersomes embedded on the surface of film appeared round in shape. It could be seen that most transfersomes were confined in the film. The shape of the crystals was changed from plane lamellar to circular shape in form of transfersomes. The SEM results showed that the transfersomes in TOFs successfully converted EBT crystals to amorphous form.

### 3.6. Atomic Force Microscopy (AFM)

The height of films from different points were used to describe the morphological characteristics and roughness of the developed TOFs. The calculated average height of film was 271.19 ± 11.7 nm, after measuring from three different points. The slight fluctuation in the height of film could be attributed to the presence of transfersomes and different components in the film; however, the overall film surface was considered smooth. The AFM results confirmed incorporated transfersomes sizes to be in the nano range in TOFs. The penetration of transfersomes in film can be visualized in Figure 5. Moreover, the scanning of film was performed horizontally in lift mode to obtain deflection view. Attractive forces were found between transfersomes and polymers of the film that embedded the transfersomes in the film. The small particle appearance in the deflection image was representative of the size of transfersomes based on force–distance curves.

### 3.7. Powder X-ray Diffraction (PXRD)

The spectra of pure drug revealed numerous diffraction peaks with high intensities at 16.6°, 18.8° and 19.4°, reflecting its crystalline nature (Figure 6A). On the other hand, XRD patterns of TOFs were completely different from those of pure drug, which might be ascribed to its conversion into amorphous form in transfersomes (Figure 6B). In the ETF-5 diffractogram, a few minor peaks were found that may be attributed to film polymers. The absence of large ETF-5 diffractogram peaks, such as 16.6°, 18.8° and 19.4°, indicates that either EBT was converted into amorphous form or completely enclosed in HPMC film.

### 3.8. Differential Scanning Calorimetry (DSC)

In DSC measurements, the pure drug exhibited a sharp melting endothermic peak at 82 °C, which was not present in the thermogram of the optimized formulation (i.e., ETF-5), indicating the conversion of crystalline drug into amorphous form (Figure 7). The components of transfersomes and oral films played a role in the conversion of the crystal structure of EBT into amorphous form. In ETF-5, the melting endotherm was invisible, which indicated amorphous drug containment in the transfersomes and HPMC film. In addition, the ETF-5 thermogram showed no sign of a dehydration peak, glass transition temperature (Tg) peak and endothermic peak, which supports the non-crystalline state of EBT. 

### 3.9. In Vitro Drug Release Studies

A significant difference was observed in statistical calculations of release pattern of the transfersomal formulations (*p* < 0.05). The drug release from all the formulations followed zero order kinetics (*r*^2^ = 0.98). The drug release from ETF-5 followed controlled release as the value of the release exponent (*n*) was greater than 1, while EBT plain film followed the Fickian diffusion mechanism (*n* < 0.5) in the Korsmeyer–Peppas model. The rate of dissolution from TOFs was significantly increased, in contrast to the pure drug (Figure 8) (p < 0.05). For instance, only 37 ± 1.8% of drug was released in 6 h and around 48 ± 3.1% in 24 h from crystalline drug suspension. In comparison, more than 52 ± 2.4% of drug was released in the first 6 h and about 82 ± 1.5% in 24 h from ETF-5 formulation. In comparison, drug release from plain film (loaded with crystalline drug) was greater than that of pure drug. The order of cumulative drug release was found as EBT < plain film < ETF-5 (*p* < 0.05). ETF-5 greatly improved the release profile of EBT relative to pure EBT and plain film.

### 3.10. Ex Vivo Permeation Studies

The cumulative drug permeated through membrane from EBT suspension was found to be 21.93 ± 3.67%, while it was 34.55 ± 2.84% from EBT-HPMC-K15M plain film (*p* < 0.05). In addition, the cumulative EBT permeated through membrane from ETF-5 was 62.82 ± 1.53% (*p* < 0.05). The cumulative drug permeated from the ETF-5 was expressively greater relative to plain film (*p* < 0.05). The amount of drug permeated by ETF-5 was 2.86-fold higher than pure drug and 1.81-fold higher than plain film (Figure 9) (*p* < 0.05). Furthermore, the highest permeation flux obtained was 1.96 μg/cm^2^/h from ETF-5. The order of drug permeation through membrane was of the following order: ETF-5 > plain film > EBT suspension. In addition, the model dependent kinetic method was applied to the permeability results of the EBT suspension, plain film and TOFs. The results found were a good fit to zero order kinetics (*R*^2^ 0.997). The highest correlation coefficient (*R*^2^ > 0.973) was obtained through the Higuchi model for pure suspension, plain film and TOFs. The n values of Korsmeyer–Peppas model showed that plain film followed Fickian (Quasi) diffusion (*n* < 0.45). TOFs followed non-Fickian (anomalous) diffusion (*n* = 0.89).

### 3.11. In Vivo Pharmacokinetics

The EBT concentration was determined in the plasma of rats after single oral dose administration. EBT was converted to its active metabolite carebastine through extensive first-pass metabolism. The mean plasma concentration of carebastine after the administration of pure drug, plain film and ETF-5 are displayed in Figure 10. Oral EBT suspension exhibited C_max_ after 4.6 h, whereas C_max_ of plain film appeared after 4.1 h (Table 4). In contrast, the ETF-5 showed peak plasma concentration after 7.8 h. The ETF-5 exhibited extended drug absorption, compared to pure drug. The T_max_ of pure drug was smaller than that of both films (*p* < 0.05). The area under curve (AUC) of ETF-5 was significantly higher than those of oral suspension (*p* < 0.05) and the plain film. The results of the AUC were statistically significant, based on differences among comparative products.

### 3.12. In Vivo Pharmacodynamics

Table 5 shows the measurement of wheal at four different time points. ETF-5 exhibited greater wheal suppression, as compared to placebo and pure drug (EBT). The suppression efficiency of the developed TOFs was much better than that of pure EBT after a single dose. ETF-5 suppressed the wheal and flare completely within 24 h. Overall, antihistamine activity of ETF-5 was found superior with reference to all other tested products (*p* < 0.05). The disappearance of wheal and flare occurred with all treatments except placebo.

## 4. Discussion

A variety of studies were carried out to finalize the appropriate transfersomal formulation by varying the concentration of lipids and edge activators. The particle size, size distribution, entrapment efficiency and deformability were opted as selection criteria for the optimized transfersomal formulation [50]. A thin layer hydration method was found suitable for the development of transfersomes. Further, it was found from the results, that the nature of lipid, type of edge activator and manufacturing method played a critical role in the size reduction of transfersomes. The small size of transfersomes allows them to infiltrate membrane pores to increase bioavailability. The large multilamellar vesicles can be reduced to unilamellar vesicles (transfersomes) by probe sonication. The small size of transfersomes may be attributed to the amphiphilic nature of PC and EA as surfactants [51]. The PDI data expressed narrow particles distribution in the prepared formulation indicating a monodisperse population system (PDI < 0.1) [52]. The small size of transfersomes also increases the surface area of drug for diffusion. The negative zeta-potential could be due to the presence of PC in the formulations. In addition, the nonionic edge activators have nominal role in development of negative charge [53]. In addition, the concentration of phosphatidylcholine and EAs play a vital role in drug loading and entrapment efficiency [54]. As the concentration of PCs increases, the high EE is achieved [55]. The concentration of PC determines the entrapment efficiency of drug in transfersomes [56]. The EE of VS-1 and VT-1 was greater than rest of the formulations. Moreover, the deformability of transfersomes is attributed to EAs inculcated in vesicle bilayers [15]. The type of EA plays a dynamic role in the deformity and release properties of transfersomes [57]. Different edge activators (EA) (Polyoxyethylene sorbitan monooleate or sorbitan monolaurate) have different deformability properties. The results revealed that sorbitan monolaurate exhibited a better outcome in relative deformability of transfersomes than polysorbate-80. The HLB value of sorbitan monolaurate-20 may be the reason for high lipophilicity, increasing deformability efficiency of lipid bilayers [58]. The maximum deformability of transfersomes was found with VS-3 [59].

TOFs exhibited good mechanical properties which could be due to the longer chain length of HPMC [60]. Clearly, the tensile strength, flexibility, elongation, thickness and weight of film are linked to film former polymer and plasticizer [32]. The physicochemical properties of film are directly associated with the release behavior of drug from films [61]. The DSC thermograms showed the sharp melting point of EBT to be around 82°C, which was not seen in the thermograms of the optimized formulation (ETF-5), evidencing its conversion into amorphous form [62]. In addition, the flat peak confirms that the drug was well entrapped in the lipid bilayers. This was additionally confirmed by XRD data. Furthermore, no major shift in peaks was observed in the FTIR spectra, indicating stable entrapment of the drug in transfersomes. AFM morphological data identified smooth and uniform surface of the films containing transfersomes. The results obtained from the re-constitution of TOFs were similar to the original results; it means transfersomes were successfully developed upon re-dispersity of films.

In vitro drug release from TOFs was sustained release which might be attributed to the bilayer of phosphatidylcholine and HPMC-K15M matrix. The drug release was retarded more in sorbitan monolaurate transfersomal formulations than in Tween^®^-80 ones [63]. Further, the increase in the permeation flux is attributed to vesicle size and deformability property, which was largely influenced by sorbitan monolaurate. The permeation improvement can be due to small size transfersomes that can penetrate the pores of the membranes easily. In addition, the presence of EA increases the permeability by fluidizing lipid bilayers [64]. Thus, drug permeability through membranes may be attributed to size, lipophilicity and deformability of transfersomes [28].

After swallowing, films were partially degraded in the stomach and almost completely in the intestine. The transferosomes were released from the films and penetrated through the GIT membranes. A small quantity of EBT was released in the stomach, which is attributed to slow erosion and gel formation of HPMC-K15. Nevertheless, major drug release occurred in the intestine due to the dissolution of TOFs and high absorption of transfersomes. Transfersomes encapsulated EBT, thus drug release from transfersomes followed a sustain release pattern. The transfersomes in developed films were found very capable to cross the transmucosal tissues barriers, due to their ability to penetrate into deeper tissues along with the therapeutic agent to enhance bioavailability. The increased permeability of EBT confirmed the advantage of a combinational approach of transfersomes and oral film [65]. Transfersomes revealed improved and controlled release of EBT. The release of drug from plain film was governed by diffusion and erosion mechanisms, whereas TOFs followed a diffusion permeation mechanism. The concentration of EBT in plasma was not detected, thus the clinically relevant active metabolite carebastine was measured [66,67]. TOFs also have a significant impact on increasing the bioavailability of EBT, as depicted from pharmacokinetic findings of carebastine in in vivo studies. The increase in T_max_ could be related to the sustained drug release behavior of transfersomes. The increase in T_max_ revealed that absorption had taken place for a longer time. The high value of C_max_ was possibly due to high absorption through transmucosal membranes. The area under curve (AUC) was increased significantly with developed ETF-5 oral film. The oral bioavailability of ETF-5 was increased 2.95 times, compared to pure EBT. Likewise, oral bioavailability of ETF-5 was improved 1.7 times, in comparison with plain film. The increase in the oral absorption rate might be linked to the deformability of transfersomes in films [68]. The small size of transfersomes increases the surface of drug molecules, subsequently maintaining the surface area in oral film. The released transfersomes enhanced the absorption through the mucosal membranes [69]. All these factors collectively led to increased bioavailability of EBT [70]. Previously, the transfersomal strategy consisted in delivering the therapeutic molecules through transdermal route [71,72]. Hence, in this novel work, the bioavailability of EBT was successfully enhanced through transfersomes-loaded oral films.

The antihistaminic effect of developed films loaded with transfersomes was strikingly superior to pure EBT [73]. The histamine-induced wheal was successfully suppressed by TOFs. Importantly, the pharmacodynamics showed superior efficiency of TOFs compared to plain film. This antihistaminic efficiency may be due to the presence of transfersomes in oral films. Overall, TOFs could be an effective delivery system to improve the bioavailability of EBT for the treatment of allergic rhinitis.

## 5. Conclusions

Transfersomes-loaded oral films (TOFs) were successfully developed as a carrier for transport of EBT. The developed transfersomes were ultra-deformable and flexible vesicles that penetrated significantly through mucosal membranes for enhancing the bioavailability of encapsulated EBT. The characteristic deformability property of transfersomes to pass through membrane pores was provided by edge activators in the prepared formulations. The permeability of transfersomes was attributed to sorbitan monolaurate which enriched the deformability of transfersomes. The higher bioavailability of EBT was achieved with TOFs than with plain film. The results obtained through entrapment—efficiency, deformability, in vitro drug release, ex vivo permeation, relative oral bioavailability—support the novel carrier system (TOFs) as an improvement in the treatment of allergic rhinitis. Convincingly, the developed TOFs could be an effective carrier system for the delivery of poorly soluble EBT.

## Figures and Tables

**Figure 1 pharmaceutics-13-01315-f001:**
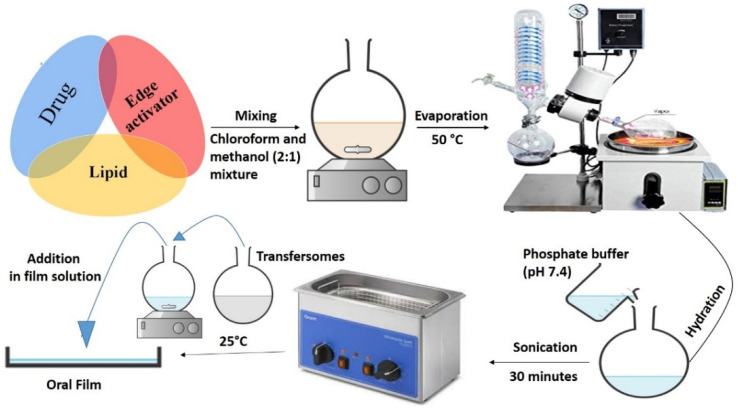
Schematic representation of the transfersomes formulation process.

**Figure 2 pharmaceutics-13-01315-f002:**
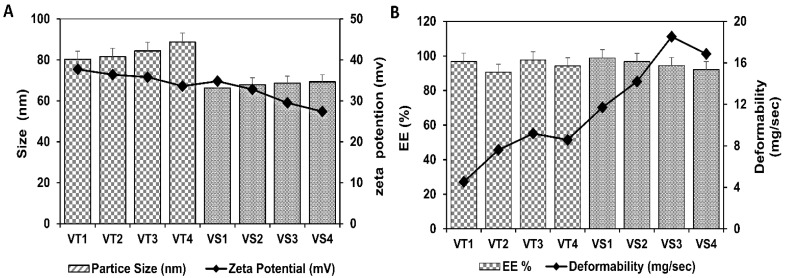
Transfersomes (**A**) size and charge, (**B**) entrapment efficiency and deformability.

**Figure 3 pharmaceutics-13-01315-f003:**
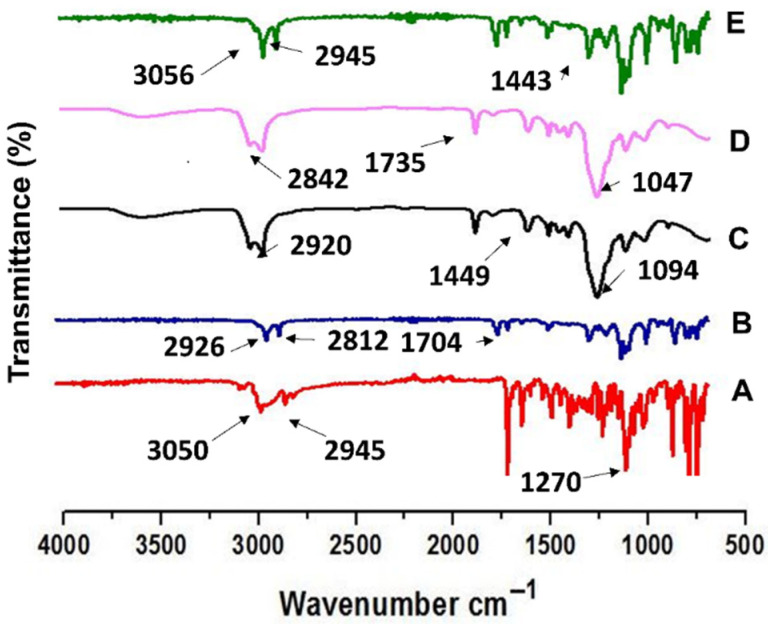
FTIR spectra of (**A**) EBT, (**B**) PC, (**C**) Tween 80^®^, (**D**) Span 20^®^ and (**E**) optimized formulation ETF-5.

**Figure 4 pharmaceutics-13-01315-f004:**
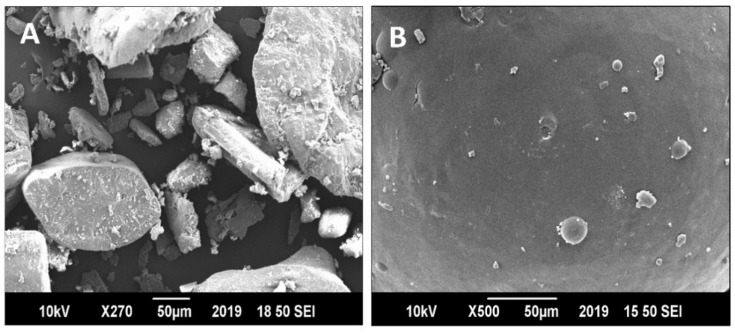
Scanning electron microscopy photographs of (**A**) EBT and (**B**) ETF-5 film.

**Figure 5 pharmaceutics-13-01315-f005:**
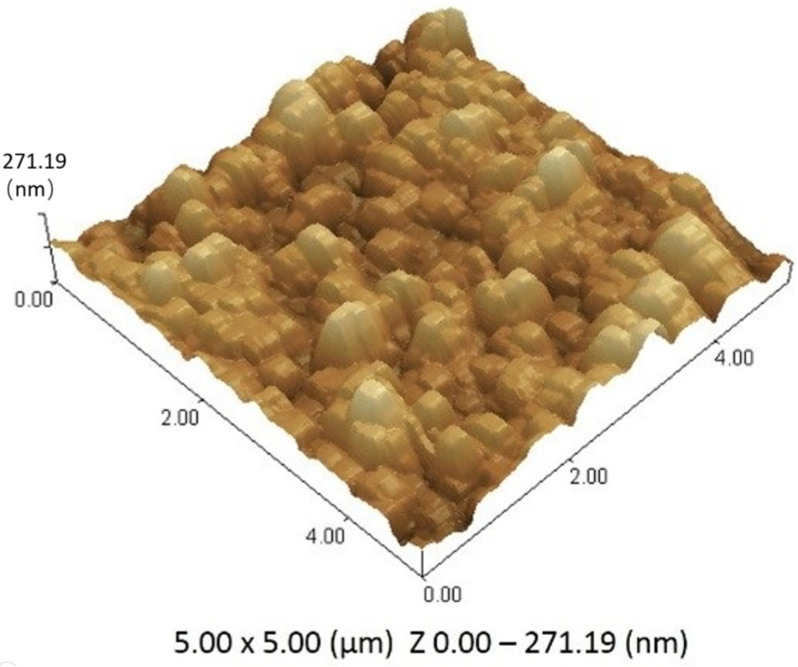
Illustration of surface morphology of optimized formulation (ETF-5) of transfersomes-loaded oral films (TOFs).

**Figure 6 pharmaceutics-13-01315-f006:**
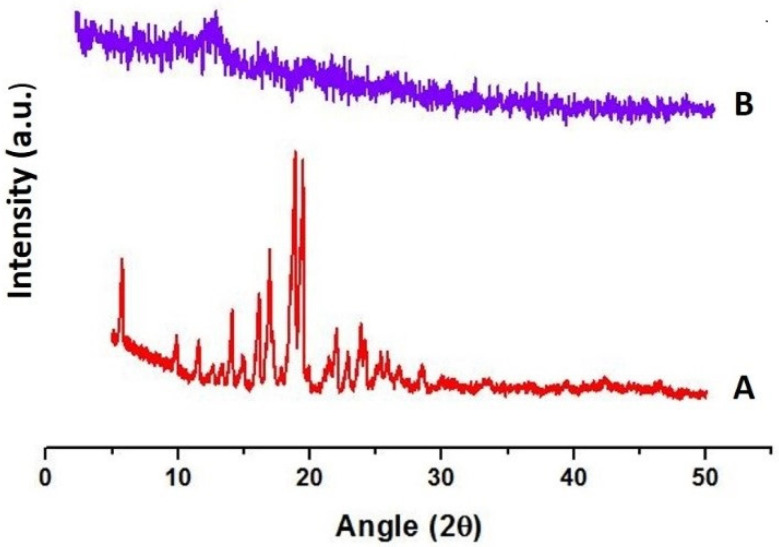
Powder X-ray diffraction of (**A**) EBT and optimized formulation (**B**) ETF-5.

**Figure 7 pharmaceutics-13-01315-f007:**
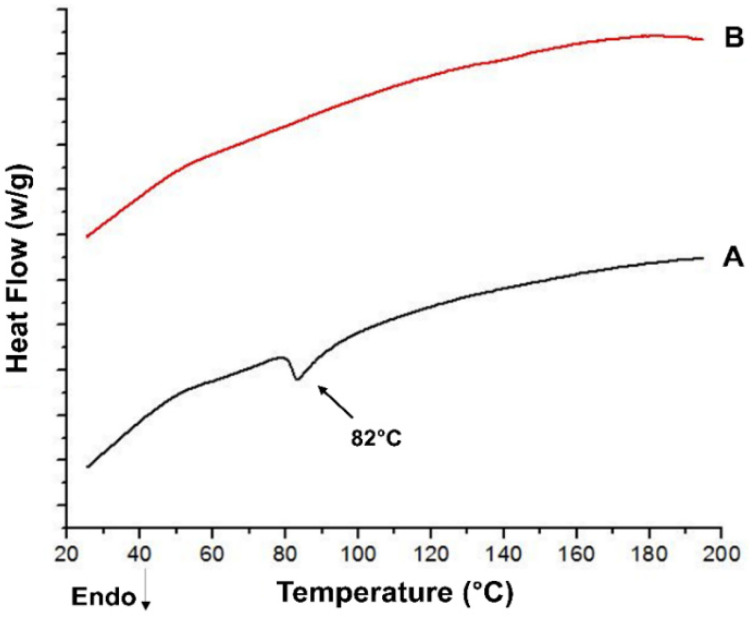
DSC thermograms of (**A**) EBT and (**B**) oral film loaded with optimized transfersomes (ETF-5-TOFs).

**Figure 8 pharmaceutics-13-01315-f008:**
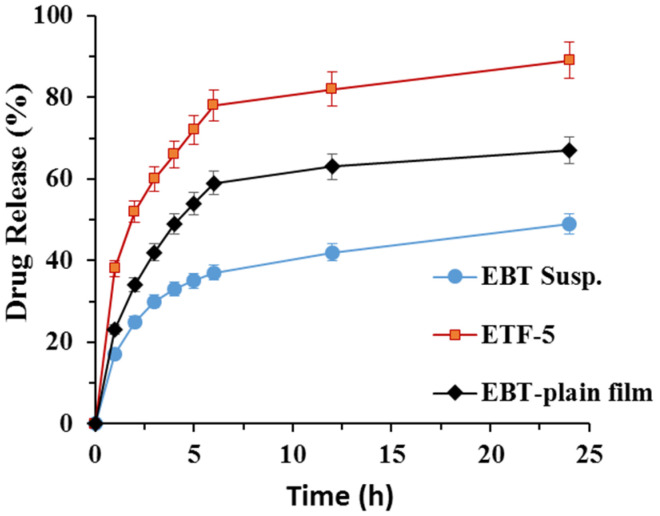
Cumulative in vitro drug release (%) from EBT suspension, EBT plain film and optimized formulation ETF-5.

**Figure 9 pharmaceutics-13-01315-f009:**
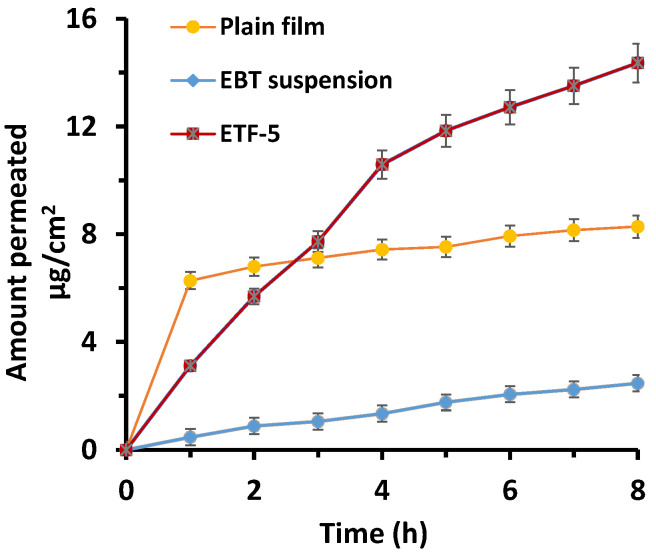
Ex vivo permeation studies of EBT suspension, EBT plain film and ETF-5 formulation.

**Figure 10 pharmaceutics-13-01315-f010:**
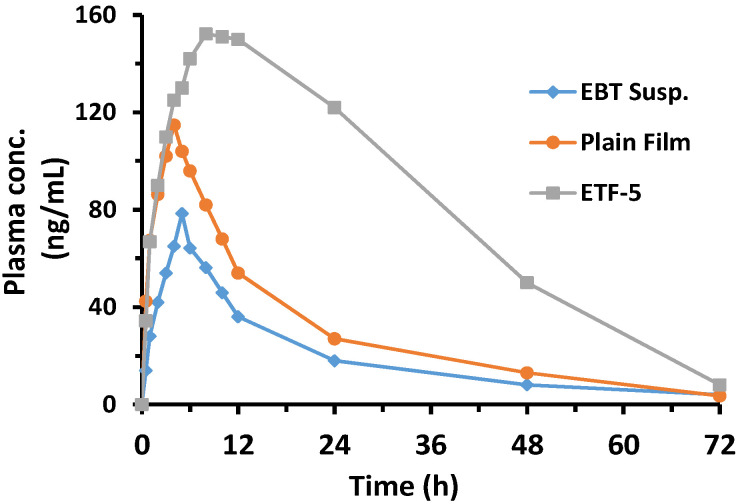
Mean plasma concentration-time profile for carebastine following oral treatment of EBT suspension, EBT plain film and ETF5.

**Table 1 pharmaceutics-13-01315-t001:** Composition of transfersomes formulations. All the formulations contained 10% *w*/*w* EBT.

Formulation Code	PC (%)	Edge Activator (%)	
		Tween 80^®^	Span 20^®^
VT-1	95	05	-
VT-2	90	10	-
VT-3	85	15	-
VT-4	80	20	-
VS-1	95	-	05
VS-2	90	-	10
VS-3	85	-	15
VS-4	80	-	20

**Table 2 pharmaceutics-13-01315-t002:** Composition of TOFs (each film contains 10 mg/mL of EBT).

TOFs Formulation	HPMC-K15M (%)	Glycerol (%)
ETF-1	2.0	4.0
ETF-2	2.5	6.0
ETF-3	3.0	8.0
ETF-4	2.0	4.0
ETF-5	2.5	6.0
ETF-6	3.0	8.0
ETF-7	2.0	4.0
ETF-8	2.5	6.0
ETF-9	3.0	8.0

**Table 3 pharmaceutics-13-01315-t003:** Results of TOFs for thickness, weight, folding endurance, tensile strength and % elongation.

TOF Formulations	Thickness (mm) ± SD	Weight (mg) ± SD	Folding Endurance (n) ± SD	Tensile Strength (Mpa) ± SD	Elongation (%) ± SD
ETF-1	0.35 ± 0.21	57.4 ± 2.64	21 ± 2.73	43 ± 1.24	4.1 ± 0.16
ETF-2	0.25 ± 0.11	69.5 ± 2.55	27 ±1.42	56 ± 1.53	5.4 ± 0.31
ETF-3	0.26 ± 0.06	87.1 ± 3.60	29 ± 2.51	67 ± 2.29	6.8 ± 0.12
ETF-4	0.24 ± 0.24	61.4 ± 1.42	34 ± 1.18	82 ± 3.22	8.2 ± 0.26
ETF-5	0.27 ± 0.04	76.2 ± 1.27	38 ± 1.45	96 ± 1.32	9.5 ± 0.12
ETF-6	0.36 ± 0.06	88.0 ± 3.30	38 ± 4.54	107 ± 2.14	10.9 ± 0.18
ETF-7	0.34 ± 0.12	62.3 ± 1.37	39 ± 3.93	76 ± 3.51	12.7 ± 0.28
ETF-8	0.36 ± 0.34	77.9 ± 1.24	41 ± 2.67	118 ± 1.04	14.8 ± 0.21
ETF-9	0.39 ± 0.26	92.1 ± 1.15	46 ± 4.88	137 ± 0.29	17.9 ± 0.31

**Table 4 pharmaceutics-13-01315-t004:** Summary of in vivo pharmacokinetic parameters of carebastine following the administration of EBT suspension, EBT plain film and EFT-5.

Pharmacokinetic Parameters	EBT Suspension	Plain Film	ETF-5 Transferosomal Film
	Group E	Group P	Group T
C_max_ (ng/mL)	78.4 ± 2.31	114.8 ± 4.01	152.3 ± 2.18
T_max_ (h)	4.6	4.1	7.8
t_1/2_ (h)	19.12	18.64	22.6
AUC_0–72_ (ng/mL/h)	2116 ± 32.51	3697 ± 12.04	6249 ± 21.7
K_e_ (1/h)	0.0372	0.0382	0.028

**Table 5 pharmaceutics-13-01315-t005:** Wheal area measurement for antihistaminic efficiency of pure EBT and developed TBF (ETF-5).

Wheal Area (mm^2^)				
Group	0 h (Baseline)	4 h	8 h	24 h
Placebo	215.4 ± 21.4	221.2 ± 18.6	224.1 ± 71.5	193.4 ± 13.2
Pure drug	219.5 ± 18.5	114.5 ± 12.1	90.43 ± 21.2	55.3 ± 3.14
ETF-5	211.8 ± 11.8	62.12 ± 10.4	12.42 ± 1.4	2.84 ± 1.0

## Data Availability

The data presented in this study are available on request from the corresponding author. The data are not publicly available as it was originally produced through research.

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
