# Peer review of "Improved Bioavailability of Ebastine through Development of Transfersomal Oral Films"

_pharmaceutics, 2021, doi:10.3390/pharmaceutics13081315_

Round 1

Reviewer 1 Report

Manuscript Islam et al. describes the production of an ebastine transferosome loaded mucoadhesive oral film (TOF) with pharmacokinetic and pharmacodynamic studies. The authors successfully produced several TOF forms and selected one of them for further investigations. They investigated the in vitro dissolutions from the new formula as compared to suspensions and plain film finding its dissolution rate supreme over the other formulas. The pharmacokinetic studies revealed that higher ebastine and carebastine (active metabolite of ebastine) plasma concentrations were achieved with the new TOF formula and the duration of high plasma level was also extended. The pharmacodynamic experiments revealed that the new formula elicited stronger action against the histamine induced wheal and flare reaction. The authors conclude that the new TOF carrier system may improve the efficacy of low water soluble ebastine in allergic rhinitis.

The followings must be answered or modified before the accepting the manuscript for publication

  1. All section started with number 1, the renumeration of the sections are necessary.
  2. Page 9, section 1.1.1: although the dermal injection of histamine induces a reaction that is like an allergic reaction, it is not an allergic response or symptom. Authors must modify the description.
  3. Page 10, Figure 2.: The Figure shows why the VS3 transferosomal formulation has been selected for further studies as compared with other formulas. However, there are other formulas with the same particle size (there is a typo in the legend of Figure 2A), zeta potential, entrapment efficiency or even deformability. For instance, the formulation VS4 seems to be not inferior as compared to SV3 considering all investigated parameters. Additionally, it is not true that SV3 would have the smallest particle size (written in the top of page 10…). Authors must clarify why only SV3 has been chosen for further studies.
  4. Page 14, Figure 8: There is a typo in the figure legend (EFT instead of ETF)
  5. Page 16: There are plasma curves for ebastine metabolite carebastine after administration of different formulas (Figure 10), but no plasma curve has been shown with ebastine, only the main pharmacokinetic parameters are shown in Table 4. It would be important to see the whole plasma curve of ebastine, and the calculated values for carebastine, thus authors must complete this part with one extra Table and Figure each.
  6. The Discussion section must be reduced avoiding the unnecessary refraining of the data from results.

Author Response

Dear Reviewer

Sir please see the attached file.

Reviewer 2 Report

This work deals with the improvement of bioavailability of Ebastine, a histamine H1 receptor blocker used to treat allergic rhinitis, obtained with a formulation constituted by transfersomes loaded in so-called oral films.

The work has some problems.

  • The aim is the preparation of “oral films for the bioavailability enhancement of Ebastine”. As reported by the authors the problems of bioavailability of this drug are due to a)GIT degradation and b) low solubility. It is not clear how the formulation they propose (oral film containing transferosome) can solve these problems? Which is the rationale of choosing such a formulation?
  • The text is confusing owing to a semantic problem. I think you use the term “oral” in an inappropriate way. For example, in the Introduction you wrote “researchers are also exploring alternative routes to avoid systematic degradation of drugs in GIT. In this regard, oral route often offers relatively permeable membrane to enhance absorptions of drug molecules compared to the alternative routes [12].” Ref. 12 is a paper in which it is described a “buccal” formulation, not an “oral” formulation. “Oral formulation” means you swallow it.
  • If you intend a formulation that is swallowed why mucoadhesion tests? Why ex vivo permeation tests?
  • If you intend a buccal mucoadhesive film to be applied to buccal mucosa why the “In vivo pharmacokinetic study” in which “samples were administered orally to all animals of three groups.”?
  • The authors report that Ebastine has low aqueous solubility: which is water solubility of this drug?
  • In vitro drug release tests were carried out in 10 mL of PBS (pH 7.4). Which is the solubility of the drug in this dissolution medium? Were the tests carried out in sink conditions? Why a test directly in this pH and no test at acidic pH and why the tests were carried out only at pH 7.4? What about the behaviour of the formulations at acidic pH?
  • Paragraph “In vitro drug release studies” RESULTS: Korsmeyer-Peppas model was used for the study of in vitro drug release from the oral films. I think that the application of this model is useful for the study of drug release from hydrophilic matrices (which are tablets). The information given in case of a thin film is really limited, and I am not sure that it could have a real meaning.
  • All the paragraphs in the RESULTS Section are numbered as 1.1

Author Response

Dear Reviewer

Sir, Please see the attachment.

Reviewer 3 Report

The manuscript presented by N. Isalm at all. is an interesting paper presenting a detailed study regarding transfersomal oral films. The paper is logically written and reveals useful information and results. 

Nevertheless before publication some corrections have to be done. Thus, the paper should be carefully checked for English errors. Similarly the number of chapters and subchapters (now appear always 1, or 1.1, or 1.1.1.). Also the authors have to write what means every abbreviations at least when the abbreviation appears the first time.

Eq. 3 should be checked, the time is missing. In addition I suggest to the authors that the order "n" is add in table regarding the kinetics or in different tables, in order to better underligne  the differences between cases.

Author Response

(The authors gave the same response as above.)

Round 2

Reviewer 2 Report

The authors partially modified the original version, according to the suggestions given.

Author Response

Dear Sir

Please see the attached file. The changes have been made as per your kind suggestions.

Best Regards,

Dr. Irfan

This manuscript is a resubmission of an earlier submission. The following is a list of the peer review reports and author responses from that submission.